# *De Novo* Transcriptome Assembly and Analysis of Longevity Genes Using Subterranean Termite (*Reticulitermes chinensis*) Castes

**DOI:** 10.3390/ijms232113660

**Published:** 2022-11-07

**Authors:** Yu-Xin Li, Chen-Xu Ye, Jian Su, Ghulam Nabi, Xiao-Hong Su, Lian-Xi Xing

**Affiliations:** 1College of Life Sciences, Northwest University, No. 229, North Taibai Rd., Xi’an 710069, China; 2Institute of Nature Conservation, Polish Academy of Sciences, 31120 Krakow, Poland; 3Shaanxi Key Laboratory for Animal Conservation, Northwest University, Xi’an 710069, China; 4Key Laboratory of Resource Biology and Biotechnology in Western China, Ministry of Education, Northwest University, Xi’an 710069, China

**Keywords:** aging, *Reticulitermes chinensis*, insulin signaling pathway, RNA-sequencing, RT-qPCRs

## Abstract

The longevity phenomenon is entirely controlled by the insulin signaling pathway (IIS-pathway). Both vertebrates and invertebrates have IIS-pathways that are comparable to one another, though no one has previously described de novo transcriptome assembly of IIS-pathway-associated genes in termites. In this research, we analyzed the transcriptomes of both reproductive (primary kings “PK” and queens “PQ”, secondary worker reproductive kings “SWRK” and queens “SWRQ”) and non-reproductive (male “WM” and female “WF” workers) castes of the subterranean termite *Reticulitermes chinensis*. The goal was to identify the genes responsible for longevity in the reproductive and non-reproductive castes. Through transcriptome analysis, we annotated 103,589,264 sequence reads and 184,436 (7G) unigenes were assembled, GC performance was measured at 43.02%, and 64,046 sequences were reported as CDs sequences. Of which 35 IIS-pathway-associated genes were identified, among 35 genes, we focused on the phosphoinositide-dependent kinase-1 (*Pdk1*), protein kinase B2 (*akt2-a*), tuberous sclerosis-2 (*Tsc2*), mammalian target of rapamycin (*mTOR*), eukaryotic translation initiation factor 4E (*EIF4E*) and ribosomal protein S6 (*RPS6*) genes. Previously these genes (*Pdk1, akt2-a, mTOR*, *EIF4E*, and *RPS6*) were investigated in various organisms, that regulate physiological effects, growth factors, protein translation, cell survival, proliferation, protein synthesis, cell metabolism and survival, autophagy, fecundity rate, egg size, and follicle number, although the critical reason for longevity is still unclear in the termite castes. However, based on transcriptome profiling, the IIS-pathway-associated genes could prolong the reproductive caste lifespan and health span. Therefore, the transcriptomic shreds of evidence related to IIS-pathway genes provide new insights into the maintenance and relationships between biomolecular homeostasis and remarkable longevity. Finally, we propose a strategy for future research to decrypt the hidden costs associated with termite aging in reproductive and non-reproductive castes.

## 1. Introduction

The insulin signaling pathway (IIS-pathway) comprises all the proteins and components involved in the action of insulin within the body [1,2,3]. Insulin is the most potent physiological anabolic agent that has ever been discovered (1921) [4,5]. It is responsible for storing and synthesizing lipids, protein, and carbohydrates, and it prevents the breakdown of these macromolecules and their release into circulation [6,7]. The IIS-pathway has been thoroughly investigated in different organisms to regulate their life span [8,9]. The IIS-pathway is responsible for mediating and transducing signals across cell membranes and regulating the physiological aspects of reproduction [10,11,12].

The inhibition of upstream IIS-pathway components (daf-2 and age-1) significantly prolongs the life and health span of *Caenorhabditis elegans* [13]. On the other hand, IGF-I and IGF-II monitor growth and growth factors (GF) [13]; nutrient sensing, stress replication [14], cell functions, and metabolic regulation is gradually deteriorating due to aging [14,15]. Similarly, IIS-related genes are also responsible for activating hormones and the motility of cells [16,17].

Researchers previously reported that IIS-associated genes that prolong the lifespan of *Drosophila melanogaster* [18,19], dauer state in *C. elegans* [20,21], and reproductives diapause in *Culex pipiens* mosquito [22]. It also indicated that squirrels live for twenty-five years, rats live for three years, and *C. elegans*, *D. longispina*, and *D. melanogaster* live only for a few weeks [23,24]. Long-lived queens of insects, including *Ephemera simulans* [23], *Pogonomyrmex owyheei* [25], and *Lasius niger* can live up to ten years [26]. *Polistes canadensis* (a social insect) has a lifespan and genetic variation up to the genus level within a single colony [27]. However, there is still a significant gap between what we know and what we do not know regarding the enormous variety of aging rates among different animals and the mechanisms that might be responsible for this diversity [28]. Similarly, the royal caste of termites lives for 18–30 years, ultimately making termites an emerging model for longevity [29] and therefore requires further attention to explore its molecular pathways. However, there are no appropriate mechanisms available for determining the life span of a termite reproductive group [30,31].

We followed the IIS-pathway (PI3K-Akt are the main effector pathways in IIS-signalling) in insects [12,32,33,34] to determine the IIS-pathway in primary and secondary worker reproductives (many subterranean termite colonies consist of former worker termites or immatures that have developed into larvae and eventually become wingless reproductives to supplement the colony), and non-reproductive subterranean termites [29,35,36]. Longevity in *Reticulitermes chinensis* was determined through next-generation sequencing (NGS) or transcriptomic [36]. A detailed evaluation of transcriptome sequences was examined with developmental processes, biological, and physiological changes in *R. chinensis* castes [29]. The de novo transcriptome assembly of *Reticulitermes chinensis*, IIS-pathway genes are enriched, contributing to termite longevity and health. In addition, the reported IIS-pathway and longevity genes in secondary worker reproductive kings (SWRK) and queens (SWRQ), primary kings (PK) and queens (PK), and male (WM) and female (WF) workers were analyzed to determine their life extension mechanism and co-evolutionary process. To better understand the genetic basis for longevity, healthy lifespan, and growth regulation in social insects and other insects, this study presents a comprehensive analysis of the IIS-pathway-associated genes implicated in *R. chinensis* longevity.

## 2. Results

### 2.1. Illumina Data Sequencing and De Novo Transcriptome Assembly

RNA-seq libraries were established from *R. chinensis* caste (PK, PQ, SWRQ, SWRK, WM, and WF). We annotated a total of 103,589,264 assembled sequence reads with an average length of 561 bp and a minimum of 201 bp (Appendix A) via the Illumina HiSeqTM 4000 platform (Appendix A). The total number of CDs sequences was reported as 64,046 sequences. Through the Trinity system (trinityrnaseq r2012-04-27), 184,436 unigenes were assembled from 201 to 43,214 bp length from precise transcriptome data (7G) (Appendix A). The assembly pipeline of Trinity is made up of six castes that come one after the other: PK (PK-1, PK-2, PK-3), PQ (PQ-1, PQ-2, PQ-3), SWRK (SWRK-1, SWRK-2, SWRK-3), SWRQ (SWRQ-1, SWRQ-2, SWRQ-3), WM (WM-1, WM-2, WM-3), WF (WF-1, WF-2, WF-3). The GC content was estimated at 43.02%, suggesting that sequence quality and development are adequate for homogeneous studies [37].

### 2.2. Functional Annotation of R. chinensis

The Non-redundant (NR) protein database was used for *R. chinensis* annotation applied for BLASTx (http://www.ncbi.nlm.nih.gov/BLAST/ (accessed on 15 March 2021)) program. A total of 184,436 unigenes were annotated with an e-threshold value of 1 × 10^−5^ (Appendix A). In total, the NR database (34.44%), the Kyoto Genes and Genomes Encyclopedia (KEGG) database (23.64%), Swiss-Prot (17.22%), and the Clusters of Orthologous Groups (COG) database (15%) had unigenes which exhibited influential matches. The Pie chart designated excellent insect genome hits and marked the termite Zootermopsis nevadensis (42%) (Appendix A). In this study, we investigate the metabolic pathways that affect the longevity of the termite castes through transcriptome sequencing. Therefore, 400 termite flagellates have been previously reported to be absorbed and metabolized in cellulose metabolism. Reticulitermes termites rely on intestinal flagellates to digest cellulose and produce the end products (acetate, CO_2_, and H_2_) of cellulose fermentation by mixed flagellates (*Tritrichomonas* species). Additionally, several databases show their relatedness with *R. chinensis* unigenes like *Zootermopsis nevadensis* (42%), *Tritrichomonas foetus* (27%), *Trichomonas vaginalis G3* (14%), *Coptotermes formosanus* (5%), *Acanthamoeba glabripennis* (2%), and *Acanthamoeba castellanli* (2%). The genes 120,113 (65.12%) of the sequences were compiled and unable to identify due to the lack of annotations of *R. chinensis* genome and short gene sequences. These short sequences include *R. chinensis* genes, unigenes, or short fragments, principally from the untranslated region (5′ and 3′ UTRs) or non-conserved region of protein-coding transcriptomes (Appendix A).

### 2.3. COG, KEGG, and GO Ontology Classifications

In the functional COG classification with 25 categories, additionally annotated 36,681 unigenes, the general functional prediction had 5707 unigenes (the colossal group), followed by cell motility (97) and nuclear structure (93), the minuscule group (Table 1). We have mapped the unique sequences into KEGG to understand the biological pathways of *R. chinensis*, including the mitogen-activated protein kinase (MAPK) pathway and the PI3K-Akt pathway, the two pathways that are involved in insulin signaling. However, the PI3K pathway is the most prominent insect insulin signaling pathway. From total castes, 93,078 unigenes were analyzed in IIS-pathway using transcriptome sequencing, and 343 genes were allocated to KEGG pathways (Figure 1A–E).

All DEGs in this study were mapped to terms in the GO database and determined the functions of the differentially expressed genes. Among 59 GO terms, the level2 GO enrichment analysis classified genes according to their molecular function (MF), cellular components (CC), and biological process (PB). The results indicated that PK-vs-SWRK castes exhibited significantly up-regulated unigenes. The categorical presentation suggests that a maximum number of up-regulated genes are reported in BP (cellular processes), followed by CC (cell and cell part), and MF (catalytic activity) (Figure 1A). Among PQ-vs-SWRQ castes, the level2 GO terms were reported significantly as down-regulated. The maximum number of down-regulated are cellular processes at BP, followed by cell and cell part in CC and binding at MF (Figure 1B).

Furthermore, level 2 GO terms of SWRK-vs-SWRQ individuals are additionally considerably reported as down-regulated genes shown at BP (Figure 1C). The level2 GO terms for WF-vs-SWRQ and WM-vs-SWRK were significantly down-regulated. The categorical cluster indicates that the BP possessed cellular processes, followed by CC (cell and cell part) and catalytic activity at MF (Figure 1D,E).

### 2.4. DEGs Analysis, Protein-Coding Region Prediction (CDS) in SWRK, SWRQ, PQ, PK, WF, and WM

The DEGs for *R. chinensis* were calculated as up-regulated and downregulated genes through DEseq2. Furthermore, 173,990 DEGs were annotated from SWRK, SWRQ, PK, PQ, WM, and WF. Up-regulated DEGs were 48,354 (27.79%) and 125,636 (72.21%) down-regulated DEGs have been reported. The maximum up-regulated DEGs were identified in WF-vs-SWRQ 20,586 (42.57%) and WM-vs-SWRK 14,165 (29.86%). Similarly, the maximum down-regulated DEGs were identified in WF-vs-SWRQ 71,950 (58.27%) and 32,887 (26.17%) in WM-vs-SWRK castes (Figure 2). The significant DEGs were reported from PQ-vs-PK (2277 up-regulated and 832 downregulated), followed by SWRQ-vs-SWRK (55 up-regulated and 272 downregulated) and WM (13 up-regulated and 124 downregulated) (Figure 3). In DEGs with a *p*-value < 0.05, the expression levels of GO and KEGG pathways showed a significantly increased. The total number of CDs was 64,046 sequences (Appendix A). The transcriptome sequence analysis and annotations of *R. chinensis* caste provided valuable evidence for evaluating all unigenes (Figure 4A–E). Transcriptome review showed high expression genes cognate to the IIS-pathway in SWRK, SWRQ, PQ, PK, WM, and WF castes.

### 2.5. Caste-Specific Expression-Genes Analysis Related to the IIS-Pathway

A total of 35 IIS-pathway cognate genes (Figure 5) were examined during the transcriptome analysis. Of them, we selected six genes (*Pdk1*, *akt2-a*, *Tsc2*, *mTOR*, *EIF4E*, and *RPS6*) to validate the fold changes in these genes. A low level of akt2-a in reproductive castes may inhibit tumour invasion and metastasis. The Tuberous Sclerosis Complex (TSC) is a genetic condition that is inherited in an autosomal dominant manner. It is caused by a mutation in either the *TSC1* or *TSC2* gene and is characterized by the development of tumours or hamartomas in many organs. mTOR, EIF4E, and RPS6 also boost growth factors, physiological processes, cell metabolism and survival, autophagy, fecundity, egg size, and follicle numbers. Therefore, we test the RT-qPCRs analysis in each caste (PK, PQ, SWRQ, SWRK, WM, and WF) of *R. chinensis*. The average 2^−∆∆Ct^ levels in SWRK are as follows: akt2-a 0.01 (the average of three replicates value calculated by 2^−∆∆Ct^) (*p* ˂ 0.00), *RPS6* 0.03 (*p* ˂ 0.00), and *EIF4E* 0.03 (*p* ˂ 0.00) were reported significantly, followed by *Pdk1* 0.01 (*p* ˂ 0.01) and *mTOR* 0.08 (*p* ˂ 0.01). In SWRQ, the *akt2-a* 0.01 (*p* ˂ 0.00), *EIF4E* 0.03 (*p* ˂ 0.00), and *mTOR* 0.04 (*p* ˂ 0.01); followed by *RPS6* 0.07 (*p* ˂ 0.01). In PK, the average levels of *akt2-a*, 0.00 (*p* ˂ 0.00), *Pdk1* 0.02 (*p* ˂ 0.01), and *RPS6* 0.04 (*p* ˂ 0.01), followed by *EIF4E* 0.14 (*p* ≤ 0.03) were reported significantly. In PQ, the *akt2-a* 0.2 (*p* ˂ 0.01), *mTOR* 0.07 (*p* ≤ 0.03), *RPS6* 0.24 (*p* ≥ 0.05), and *EIF4E* were 0.18 (*p* ≥ 0.05) reported significantly. In WM, *akt2-a* 0.20 (*p* ≥ 0.05), while in WF *akt2-a* 0.25 (*p* ˂ 0.00) followed by *mTOR* 0.40 (*p* ˂ 0.01) and *Pdk1* 0.53 (*p* ≤ 0.04) were reported significantly (Figure 6). The expression of these genes: *mTOR* (SWRK 0.08; SWRQ 0.04), *Pdk1* (SWRK 0.01), *akt2-a* (SWRK 0.01; SWRQ 0.01), and *EIF4E* (SWRK 0.03; SWRQ 0.03) (Figure 6), was reported enormously in SWRK and SWRQ than non-reproductive (WM and WF). These results indicate that secondary worker reproductive like the king and queen have a more successful life span than non-reproductive castes. According to these findings, a relatively conserved protein in the insulin signaling system significantly delays or prevents age-related disorders and aging mechanisms, structures, and associated pathways.

## 3. Discussion

The transcriptome sequences of *R. chinensis* castes were compiled to determine the longevity-related genes that contribute to extending the lifespan of SWRK, SWRQ, PK, and PQ caste’s. Transcriptomically, we confirmed that several genes (*Pdk1*, *akt2-a*, *Tsc2*, *mTOR*, *EIF4E*, and *RPS6*) in *R. chinensis* had extended the lifespan like *D. melanogaster* [18,19] and *C. elegans* [20]. The SWRK and SWRQ live longer (18–30 years) than WM and WF (a few weeks to months) [19,33]. It is challenging to sort out and estimate the lifespan of termites. Therefore, we evaluated transcriptomic sequences with next-generation sequencing (NGS) associated with developmental, biological, and physiological changes in cells or tissues [29,38]. The transcriptome research was determined through quantitative RT-qPCR. However, the current study is the second endeavor to longevity-cognate genes of *R. chinensis* castes [29,38]. According to the transcriptome data, 35 DEGs involved in the longevity-related genes; *Pdk1*, *akt2-a*, *Tsc2*, *mTOR*, *EIF4E*, and *RPS6* are involved in the longevity of invertebrates and vertebrates. The insulin/insulin-like growth factor signaling pathway is a hormonally mediated cell-signaling pathway, that involves insulin-like peptides, transmembrane receptors of their cognate cell surface, and downstream effectors [19,33], which shreds of evidence annexing genetic and biochemical changes [39]. The PI3K-Akt signaling pathway plays a principal role in insects’ longevity [12,33,34].

Downstream Akt recruited and activated phosphorylation at S473 and T308 at the plasma membrane [40]. Permitted *Akt* controls cell survival, proliferation, and protein synthesis [41], and activates mTORC2 and Pdk1 [42,43]. In many organisms like yeast, *C. elegans*, and *D. melanogaster*, *Pdk1* is important in cell survival and evolution [44,45]. Additionally, for murine embryonic development, *Pdk1* is essential; mice without a *Pdk1* gene died at 9.5 days of the embryo and showed abnormalities in different tissues [46]. *Pdk1* hypomorphic mice have smaller cardiovascular organ volumes, and *Pdk1* conditional abstraction in muscle cells leads to heart failure and minimizes lifespan [47]. *Pdk1* has direct effectors on *Akt*, *S6K*, and *RSK*, causing embryonic stem cells to activate all three of these kinases [46]. However, *Akt*/*PKB* activations are involved in several cellular replications to magnification factor signalings, such as apoptosis bulwark, glucose transporter (GLT), and glycogen synthesis [10], while in *D. melanogaster*, it regulates lifespan, reproductive status, growth, and metabolism [48,49,50]. Moreover, *Pdk1* in neuronal IIS-pathway can influence chemotaxis and learning [50]. PI3K-Akt and FOXO proteins are also essential in *Drosophila*, *Nilaparvata lugens*, and *Sogatella furcifera* flight muscle genes [51,52]. In *C. elegans*, *Pdk1* controls insulin physiological effects, and growth factor (GF) enters the staggering dauer phase and elongates their life span when *Pdk1* becomes dormant [53]. The expression of the *Pdk1* gene through transcriptome and RT-PCRs analysis was significantly higher in SWRK than in any other castes. As a consequence of these findings, our results indicated that *Pdk1* serves the same active function in the life span regulation of *R. chinensis* termites as it does in *C. elegans*, *S. furcifera*, *Drosophila*, and *N. lugens*.

Tuberous sclerosis (TSC) and its domains (Tsc1 and Tsc2) are responsible for autosomal-dominating disorders, including epilepsy, skin, retina, heart, kidney, and the central nervous system [54,55]. A germ-line mutation in the rat homologous human Tsc2 gene makes the Eker rat prone to many tumors and death in the intermediate stage. Tetracycline-dependent conditions and overexpression of *Tsc2* inhibited the proliferation of an Eker rat-derived kidney tumor cell line [55]. Tsc1 in the *Drosophila* homolog is associated with the cell mutation study (mosaic screens) and polyploidy intended to trigger a cell switch [56]. Current RT-qPCR findings in this study showed that *Tsc2* is abundantly conserved among all the castes.

Protein kinase B and its three paralogs, *akt1*, *akt2*, and *akt3*, exhibit intensively deliberated cancer and metabolism [41], and also regulate the plasma membrane [57,58] and cell size mutations in *Drosophila* [56]. *Akt* activates considerable tissue growth/size and is essential for the PTEN facility to act as a tumor suppressor in *Drosophila* [10,59]. The expression of *akt2-a* may also help longevity and remove tumor cell incursion and metastasis in reproductive castes of *R. chinensis*. *mTOR* was first identified and cloned within the budding yeast *Saccharomyces cerevisiae* [60,61]. Extracellular signaling activates the *mTORC1* and *mTORC2* network in innate immune cells [11,62,63] and regulates translation, protein synthesis, mRNA translation, anabolic cell growth, and metabolism [64]. Knock-down of *mTOR* regulates low fertility in *N. lugens* and activates vitellogenin gene (Vg) expression in *A. aegypti* [65,66], also essential for transduction alimentation during mosquito egg development [67,68,69]. *mTOR* also controls ovary size in *Drosophila* [70], which promotes the number of follicles and owns diphenic development in *Apis mellifera* [71]. The previously reported results indicate that *mTOR* increases life span, fecundity, and GF in reproductive termite castes. The same strategy (RT-PCR analysis) is used for *mTOR* in WM, WF, PK, PQ, SWRK, and SWRQ, where it shows a possibility of prolonging the lifespan of termite castes.

The *RPS6* deficiency cells did not significantly reduce global protein translation or 5′-TOP mRNAs [9]. However, functional analysis and mutations in *eIF4E-1* result in embryonic defects and lethality [72,73]. Our results suggest that IIS-related pathway genes increase reproductive castes’ life expectancy and control a metabolic pathway in *D. melanogaster*, *R. chinensis*, *C. elegans*, and fly. In addition, their nutritional reactions and sensory compensation for IIS-pathways and insulin peptides are similar [13,74,75].

In the current investigation, the hypothesized findings (RT-qPCR results) were obtained from the reproductive castes (PK, PQ, SWRK, and SWRQ), with much higher expression than in the non-reproductive castes (WM and WF). Our findings ultimately shed new insight into the maintenance and significance of biomolecular homeostasis in reproductive and nonproductive *R. chinensis* castes. As previously described by a number of researchers, IIS pathway-related genes desperately promote growth factors, physiological processes, cell metabolism and survival, autophagy, fecundity rate, egg size, and follicle number. These findings imply that a relatively conserved protein in the insulin signaling system significantly prolongs or avoids age-related diseases, processes, structures, and associated pathways. Further research is required to determine the biological function of genes connected to the insulin signaling pathway using genetic tools such as RNA interference and transgenic structures in order to determine how these genes contribute to the continued existence of termites. In the end, the findings we were able to obtain new shreds of insight into the processes involved in preserving biomolecule homeostasis and its connection to remarkable longevity.

## 4. Materials and Methods

### 4.1. Collection and Rearing of Reticulitermes Chinensis

Termite *Reticulitermes chinensis* colonies were dug from Chengdu in April–May 2014 (Table 2) and transferred to the laboratory via plastic boxes (25 cm × 18 cm × 15 cm). Isolated colonies were reared for 6 years under generous (munificent) and crowded conditions (the lab was 24 h open) at room temperature (25–28 ± 1 °C) at Northwest University (http://english.nwu.edu.cn/ (accessed on 7 June 2021), Xian, Shaanxi, China. A total of 250 monohybrid colonies were established; each colony corresponds to male × female alates 78% (195/250) and female × female alates 22% (55/250) using pine sawdust (50–60% humidity) in specially designed plastic boxes (80 mm × 65 mm × 40 mm) [76]. At an early stage, active and mature colonies were selected for the experiment, from where we accumulated *R. chinensis* reproductive (PQ and PK; SWRQ and SWRK) and non-reproductive (WF and WM) castes [29]. These collections were done randomly from different colonies (not specific to the quantities and targeted colonies), where we found the primary and secondary reproductives and non-reproductives castes. Anyhow, many factors influence the division of labour in an insect community, including the size of the colony, castes and instar demography, food availability, and competition [77]. For example, the size of the *Rhytidoponera metallica* (Smith) ant and *Coptotermes formosanus* [78] colony influences the existence of age polyethism [79]. Colonies of *R. chinensis* grow from a mated pair (30–40 days start egg laying) to an immature colony (7–25 months), a juvenile colony (3–6 years), and a sexually mature colony (after 6 years). In a single colony, each caste performs its own role (e.g., food foraging and colony carrying). Due to the fact that, during the rearing of the colonies, a variety of colonies died due to fungus attacks (paralyzing the termites till death), while some colonies laid eggs very late (an average period for eggs laying was observed from 30–40 days). According to these fundamental reasons, the termites were collected randomly.

### 4.2. Experimental Samples

During the study, *R. chinensis* colonies were subjected to synchronization procedures immediately after six years of rearing of these colonies because we were looking for selected castes, i.e., primary king (PK) and queen (PQ), secondary worker reproductive king (SWRK) and queen (SWRQ), worker male (WM) and female (WF). Total RNA was extracted from the whole body (for RT-PCR, we used heads of individuals free from protozoans) of PQ, PK, SWRQ, SWRK, WF, and WM for RNA Illumina sequencing [29]. Three technical (e.g., PK-1, PK-2, and PK-3) and five biological replicates (e.g., PK-1 had five individuals) (randomly collected from 250 colonies) of each caste (six in number “PQ, PK, SWRQ, SWRK, WF, and WM”) were used.

### 4.3. Total RNA Extraction, cDNA Synthesis, and Illumina Sequencing

The whole body of *R. chinensis* castes was micro-dissected (into the head, legs, thorax, and wing (of an adult) to extract DNA and RNA), and it was promptly stored in liquid nitrogen.

We used TRIzol reagent and an Agilent 2100 Bioanalyzer in order to collect sufficient RNA for Illumina sequencing (Agilent Technologies, Palo Alto, CA, USA). The Lysate RZ (TRIzol reagent) and RNAsimple Total RNA-Kit (Tiangen Biotech “Beijing” Co., Ltd., Beijing, China) were applied to the tissue to homogenize and prevent the tissue from degrading [76]. Tissue homogenates were heated to temperatures ranging from 15–30 °C to separate the nucleic acid-protein complex. Afterward, the tissue homogenate was transferred to RNase-free centrifuge tubes (for centrifuge spinning) to separate the aqueous phase, precipitation, deproteinized, and abstracted residual liquid. An adequate quantity (for Illumina sequencing and RT-PCR experiments) of total RNA was extracted and stored at −80 °C for further experiments. Finally, with the help of a spectrophotometer (NanoReady “Model: F-1100”, Shanghai, China), the total amount of RNA (protein: 260/280 and salt: 260/230) was checked to quantify and verify the entire amount of RNA integrity [29].

After that, the NEB-Next Prep-Kit for Illumina (NEB) sequences [80] was used to revert cDNA in accordance with the manufacturer protocol. Furthermore, we used the QiaQuick PCR extraction kit for cDNA fragments emasculated. The poly(A) end-paired were tailed and combined with the sequence of the Illumina adapter. We obtained sequence readings of an average length (561 bp) and a minimum length (201 bp) by using the Illumina HiSeqTM 4000 platform (Appendix A). The ligation product was selected by size from the Gene Denovo Biotechnology Co (Guangzhou, China). In addition, the Trinity system (trinityrnaseq r2012-04-27) was used for each test simultaneously in order to obtain an adequate number of clear unigenes [81,82].

### 4.4. Transcriptome Assembly and Reads Mapping

Raw data evaluation affects the quality and screening; therefore, we used fastp (version 0.18.0) before and after filtering and using the following parameters (˃10% of unknown and ˂50% inference of reads) for readings that contain adapters of low-quality q-value (≤20 bases) [83]. Quality clean reads of the transcriptome de novo was assembled with the short-read assembly through the trinityrnaseq r2012-04-27 assembly program [81]. Further data on the development of transcriptomic assemblies using a de Bruijn graph algorithm from short-read sequences have been provided in supplementary data [84].

### 4.5. Read Alignments, Normalization, and the Gene Expression Level

Short sequences and readings have been mapped into a reference SOAPaligner/soap2 tool [85,86]. The edgeR (Bioconductor package version 2.4.0) was used to generate read counts for genomic features and summarize short reads (edgeR is a method that is based on the weighted mean of log ratios). This method normalizes data in the beginning by calculating size and normalization factors from raw reads of the Illumina^TM^ sequence. The program implements precisely established statistical approaches for multigroup experiments [87]. These genes were tenacious by the quantifications allocated exclusively to the exon region per million mapped reads (RPKM) [76]. The R-package (http://www.r-project.org (accessed on 15 March 2021)) was used to calculate all statistical data expression and visualization [37].

### 4.6. Differentially Expressed Genes (DEGs), Identification, Validation, and Functional Enrichment Analyses

After reading alignments and normalization of genes, finally, we obtained DEGs, using DEseq2 (because each caste has three replicates); the relative expression level of each gene was calculated [88]. These statistical methods distinguish between digital gene expression data (electronic data) of a negative binomial distribution model, fold changes (FC > 2), and the false discovery rate (FDR < 0.01) to justify the threshold level (*p*-value). The value of log2ratio ≥ 1 is an absolute value in calculations of significant DEGs between pooling samples. Annotations from the Gene Ontology (GO), the Kyoto Genes and Genomes Encyclopedia (KEGG) (http://www.genome.jp/kegg (accessed on 15 March 2021), the Clusters of Orthologous Groups (COG) (https://www.ncbi.nlm.nih.gov/COG/ (accessed on 15 March 2021) and the Non-redundant (NR) (http://www.ncbi.nlm.nih.gov (accessed on 15 March 2021) database were all relegated [86]. Blast2GO software assigned the GO annotation (obtained from the Nr database profile) [89]. In addition, WEGO software was used to classify unigenes functions [90].

### 4.7. Quantitative Real-Time PCR (RT-qPCR) Assay

The RT-qPCR assay was performed in a 20 μL reaction volume SuperReal PreMix Plus (Tiangen, Sichuan, China) with CFX 96 (Bio-Rad system) following the manufacturer’s instructions. The Beta-actin gene averted fluctuations in quality and quantity for Reticulitermes termites [35,83] and was selected as a housekeeping gene [37,91]. We designed primer pairs for each IIS pathway-cognate gene using Primer3 v1.1.4 (Appendix A) [29]. For RT-qPCR, the total RNA was extracted from the head (free from protozoans) of each caste (PK, PQ, SWRQ, SWRK, WM, and WF), using an RNAsimple Total RNA kit (Tiangen). Three technical (eg., PK-1, PK-2, and PK-3) and five biological replicates (randomly collected from different colonies) were used (e.g., six castes, three replicates, and each replicate consisted of five individuals). We used Fast-King RT-Kit (Tiangen) for cDNA library construction (cDNA kept for a further experiment at −20 °C). The cDNA (4 μL), SYBR Premix-Ex Taq-TMII (10 μL), and ddH_2_O (4.8 μL) were performed in a quantitative reaction (20 μL) on life technologies/Vii7, 0.6 μL of the reverse and forward primers. RT-qPCR was amplified at 95 °C (15 min), 95 °C (10 s), 60 °C (20 s), and 72 °C (32 s). Following amplification, the melt curve (at CFX 96 Bio-Rad system) analysis detected a gene-specific and nonspecific amplification.

### 4.8. Statistical Data Analysis

A standard method 2^−∆∆Ct^ (formulae are given below) was used to calculate the relative gene expression of the mRNA genes used with each replicate [92,93]. The RT-qPCR experiment quantifying and counting results were statistically analyzed using IBM Corp. Released 2019 (IBM SPSS Statistics for Windows, Version 26.0. Armonk, NY, USA: IBM Corp.) to determine the relationship between groups (reproductive and non-reproductives) using a *t*-test [29]. All values were expressed as mean ± standard deviation (*p*-values < 0.05) and statistical figures constructed with Microsoft Excel (0365) and OriginPro (2018).
(1)$$\mathrm{Ratio}=\frac{{{\left(E_{t\mathrm{gene}\mathrm{of}\mathrm{interest}}\right)}^{\Delta \mathrm{Ct}}}_{t\mathrm{gene}\mathrm{of}\mathrm{interest}}}{{{\left(E_{\mathrm{housekeeping}\mathrm{gene}}\right)}^{\Delta \mathrm{Ct}}}_{\mathrm{housekeeping}\mathrm{gene}}}$$
whereas ${{}^{\Delta \mathrm{Ct}}}_{t\mathrm{gene}\mathrm{of}\mathrm{interest}}{{=}^{\mathrm{Ct}}}_{\mathrm{control}}{{-}^{\mathrm{Ct}}}_{\mathrm{treatment}}$ and ${{}^{\Delta \mathrm{Ct}}}_{\mathrm{housekeeping}\mathrm{gene}}{{=}^{\mathrm{Ct}}}_{\mathrm{control}}{{-}^{\mathrm{Ct}}}_{\mathrm{treatment}}$

(2)$$\mathrm{Ratio}=2^{-\Delta \Delta \mathrm{Ct}}$$
whereas ${}^{\Delta \Delta \mathrm{Ct}}{{=}^{\Delta \mathrm{Ct}}}_{\mathrm{housekeeping}\mathrm{gene}}{{-}^{\Delta \mathrm{Ct}}}_{t\mathrm{gene}\mathrm{of}\mathrm{interest}}$.

## Figures and Tables

**Figure 1 ijms-23-13660-f001:**
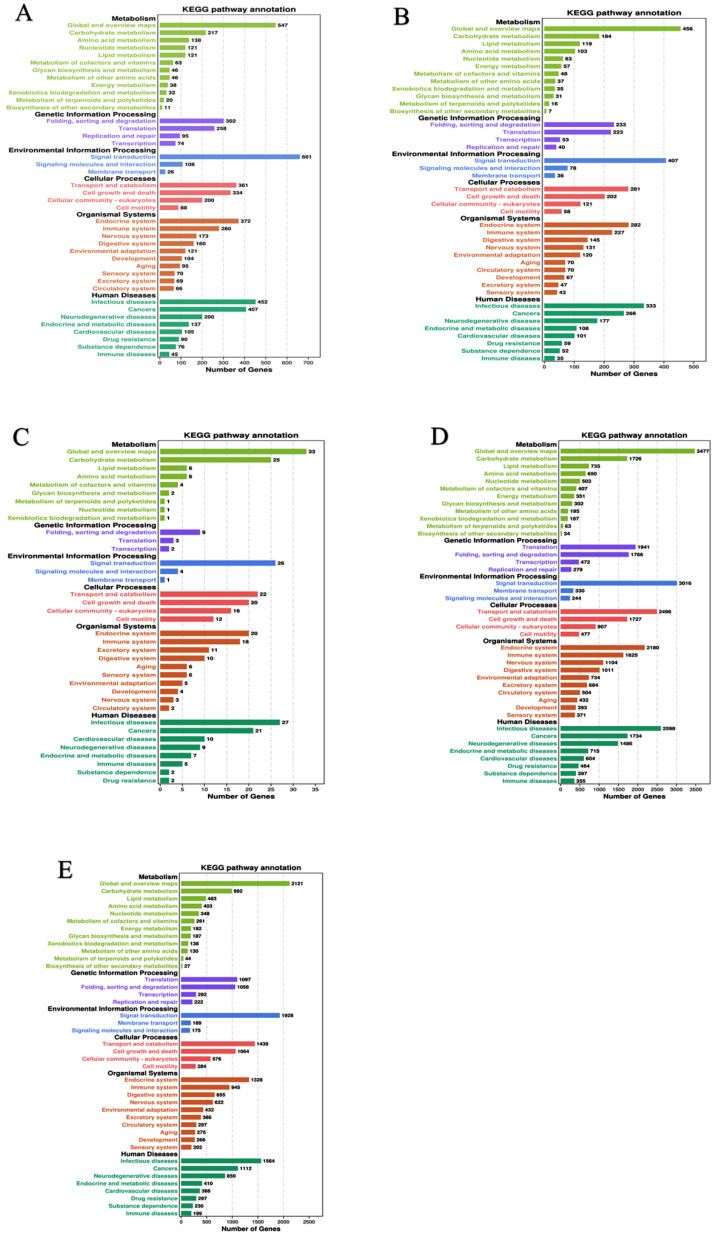
KEGG classifications of the *R. chinensis* unigenes. (**A**) PK-vs-SWRK; (**B**) PQ-vs-SWRQ; (**C**) SWRK-vs-SWRQ; (**D**) WF-vs-SWRQ; (**E**) WM-vs-SWRK. PKs (primary king), PQs (primary queen), SWRK (secondary worker reproductive king), SWRQ (secondary worker reproductive queen), WMs (workers male), and WFs (female workers), with 93,078 unigenes were assigned to 343 pathways.

**Figure 2 ijms-23-13660-f002:**
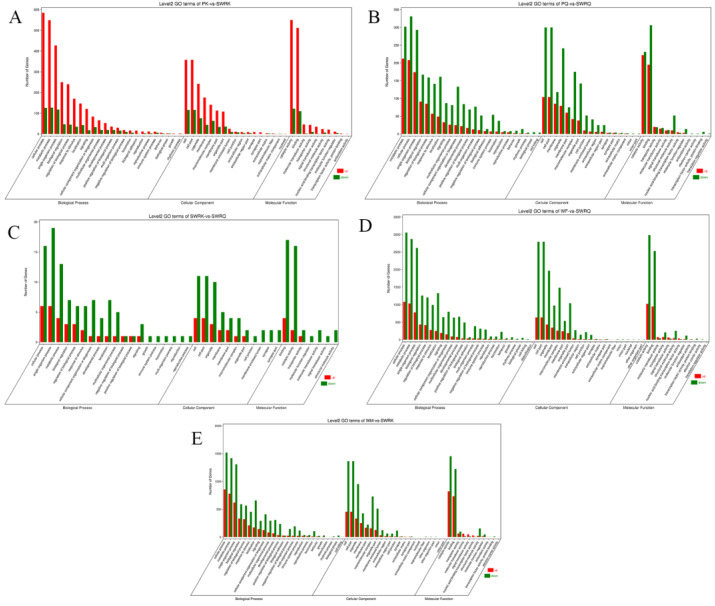
Histogram presentation of the Gene Ontology (GO) classification in each caste. (**A**) PK-vs-SWRK; (**B**) PQ-vs-SWRQ; (**C**) SWRK-vs-SWRQ; (**D**) WF-vs-SWRQ; (**E**) WM-vs-SWRK. PKs (primary king), PQs (primary queen), SWRK (secondary worker reproductive king), SWRQ (secondary worker reproductive queen), WMs (workers male), and WFs (female workers). The figure represents the up (red) and down (green) categorical presentation of biological processes, cellular components, and molecular functions. The *x*-axis indicates the names of genes in a category, and the *y*-axis shows the number of genes in the main category.

**Figure 3 ijms-23-13660-f003:**
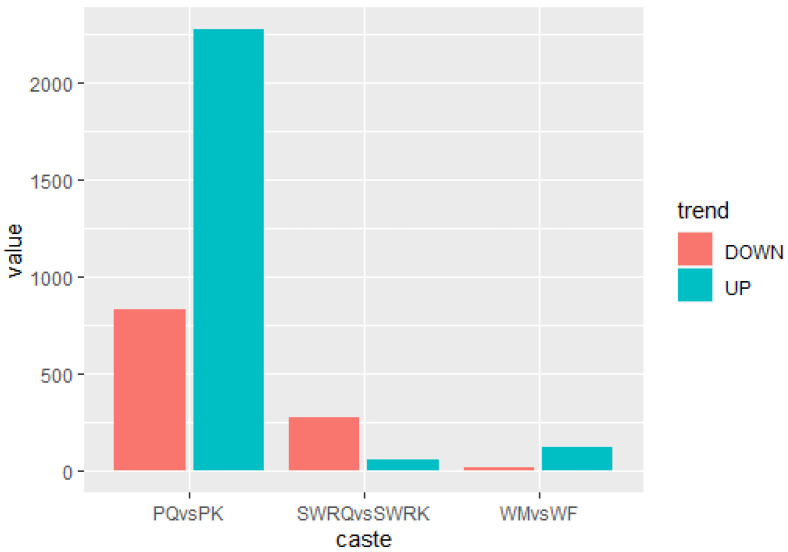
Differential expression of genes (DEGs) analysis across distinct castes of *R. chinensis* (SWRQ-vs-SWRK, PQ-vs-PK, and WM-vs-WF). The red column represents DEGs that are down-regulated, whereas the green column indicates DEGs that are up-regulated. To determine the significance of gene expression changes, FDR ≤ 0.001 and log2Ratio ≥ 1 were employed as thresholds.

**Figure 4 ijms-23-13660-f004:**
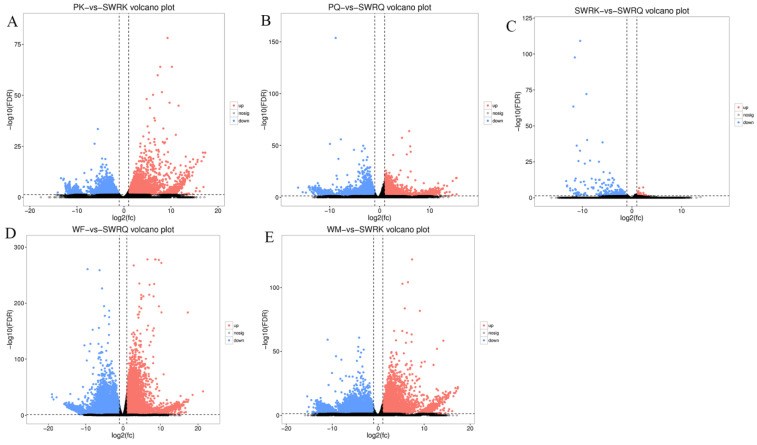
Analysis of DEGs for enhancement across different castes (**A**–**E**). PK (primary king), PQ (primary queen), SWRK (secondary worker reproductive king), SWRQ (secondary worker reproductive queen), WM (male worker), and WF (female worker). The volcano plots indicate enriched DEGs expressing genes in the PK, PQ, SWRK, SWRQ, WM, and WF, with a significantly up-regulated (red) and down-regulated (blue). The threshold level for judging the significance of differences in gene expression, FDR ≤ 0.001 and log2Ratio ≥ 1 parameter were used.

**Figure 5 ijms-23-13660-f005:**
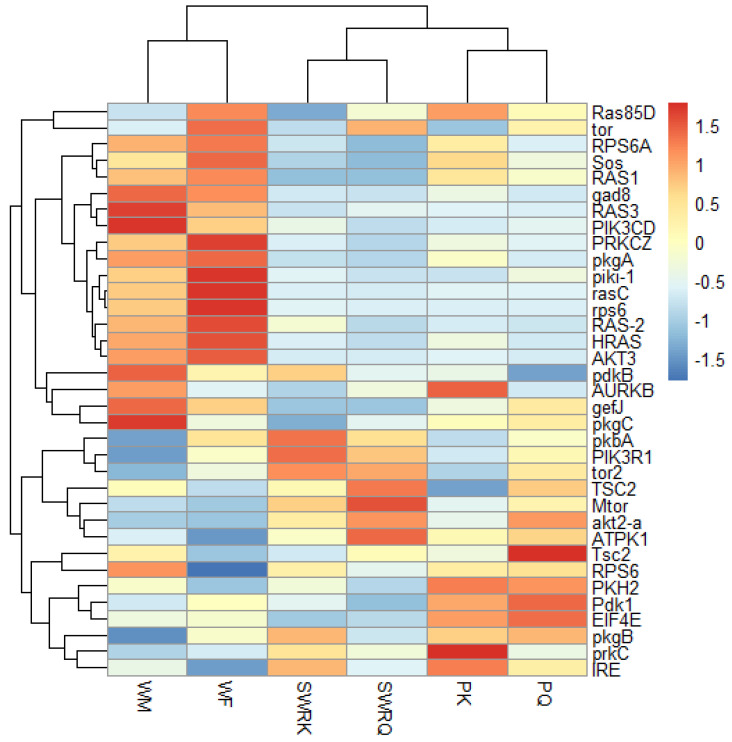
The heat map indicates the differentially expressed gene (DEGs) involved in the insulin signaling pathway in different castes of *R. chinensis*. The castes are SWRK, SWRQ, PQ, PK, WM, and WF. A change in color from red to blue indicates the gene expression level.

**Figure 6 ijms-23-13660-f006:**
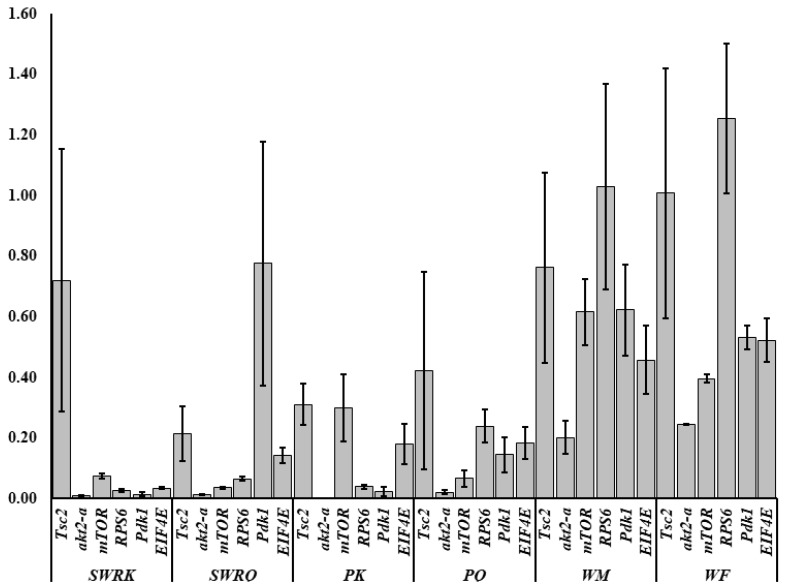
Log2 fold changes in the insulin signaling pathway in *R. chinensis* castes SWRK, SWRQ, PK, PQ, WM, and WF. Differential expression of genes in each group is shown as log2 fold changes compared to the reference group. The y-axis represents the log2 fold changes, and the x-axis shows the gene name and individual caste name.

**Table 1 ijms-23-13660-t001:** The table represents clusters of the orthologous group (COG) classifications. In 25 categories of COG classification, a total of 36,681 unigenes were recorded.

COG Classification	Number of Unigenes
General function prediction only	5707
Signal transduction mechanisms	4884
Posttranslational modification, protein turnover, chaperones	4170
Translation, ribosomal structure, and biogenesis	2632
Intracellular trafficking, secretion, and vesicular transport	2311
Cytoskeleton	2168
Carbohydrate transport and metabolism	1563
Cell cycle control, cell division, chromosome partitioning	1410
Function unknown	1409
Transcription	1404
Amino acid transport and metabolism	1125
Lipid transport and metabolism	1121
RNA processing and modification	1116
Lipid transport and metabolism	957
Energy production and conversion	940
Secondary metabolite biosynthesis, transport, and catabolism	812
Chromatin structure and dynamics	583
Replication, recombination, and repair	547
Nucleotide transport and metabolism	468
Defense mechanisms	340
Coenzyme transport and metabolism	304
Cell wall/membrane/envelope biogenesis	284
Extracellular structures	236
Cell motility	97
Nuclear structure	93

**Table 2 ijms-23-13660-t002:** The information of *R. chinensis* wild colonies and the collected number of different castes during 2014.

Colony and Location	Collection Date	Neotenic (Ne)	Work (W)	Soldier (S)	Nymph (N)	Alate (A)
1. Xingfu Road, East New District of Chengdu	7 April 2014	95	164	56	133	241
2. Oupeng Avenue, Longquanyi District	16 February 2014	25	134	37	120	163
3. Fenghuang Avenue, Qingbaijiang District	12 May 2014	-	122	12	-	95
4. Yingbin Avenue, Xinjin District	11 May 2014	64	170	110	160	130 *

- not found or not present. * The alates were collected from the same colony in April 2016.

## Data Availability

The transcriptome data will be available under the accession number (BioProject: PRJNA592596) at the NCBI database. The corresponding author will answer any question upon request.

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
