# Peer review of "De Novo Transcriptome Assembly and Analysis of Longevity Genes Using Subterranean Termite (Reticulitermes chinensis) Castes"

_ijms, 2022, doi:10.3390/ijms232113660_

Round 1

Reviewer 1 Report

In my opinion, it is necessary to carefully review the references cited throughout the article, there are many errors throughout the entire introduction and discussion.

The scientific names of the species are written in italics, correct this throughout the text.

lines 2-3 I think you should be more specific in your title because your work focused on IIS-pathway associated genes.

line 24 remove "assembled" reads refers to the raw sequences provided by the sequencer before assembly

lines 55 and 56 To specify the references corresponding to C. elegans, the article by Giannakou and Partridge, 2007 corresponds to Drosophila.

lines 85 and 375 is next-generation sequencing (NGS)

line 64 the article by Li et al., 2009 is about a bioinformatics tool.

linea 66 please specify a reference for Culex pipiens because Kennedy, 2008 is a review, and Tissenbaum, 2012 is in C. elegans.

lines 66,67 and 68 correct text or reference because Carey's 2002 study was conducted on Hexagenia limbata and Ephemera simulans, and Jemielity et al., 2005, the authors discuss studies on aging in social insects  

lines 68-70 The reference for this sentence is missing.

line 80   check the reference (Taniguchi et al., 2006)

lines 83 and 84 correct text or change references Mirzoyan et al., 2019 is a study in Drosophila.

lines 85 Next-generation sequencing (NGS) 

lines 86 the articles cited did not study the species Reticulitermes chinensis review references 

lines 101 check font size in the word munificent

line 105 The analysis females x females is in different castes?

line 115 remove the dot inside the parentheses

line 147 spinning means centrifuge?

line 228 same as line 24

linea 154, 156, 159 and 160 It is convenient to write down the catalog number of the reagents used.

lines 160-162 check the citation for this sentence or describe if the methodology was like the one used in that article.

line 174 here, it corresponds to note the reference number 40, Li et al., 2009

line 241 I suggest changing the reference to Figure S5 at the end of the paragraph

linea 252 missing letter "t" Tritrichomonas

line 271 review the total sum of unigenes reported in table 2 and correct text (36,681)

line 297 corrects the number 125,636

line 258 Please check if this comment corresponds to the figure (S5)

line 320 It is convenient to maintain the same order of the castes both in the figure and in the figure legend

lines 320 and 321 the text in the figure legend is contrary to the figure, insert the indication is inverted red and green

lines 334-337 I don't understand that Tsc2 is one of the 6 genes that they selected to analyze and two lines later they say "Tsc2 is an autosomal-dominant disease that causes epilepsy, body volume, and mutant cells"

lines 339-353 I think a table with these results is better

line 363 corrects the word "cates"

line 375 I suggest changing "or" to "with" or "from" Next-generation sequencing because the abbreviation (NGS) is used to refer to methodologies for sequencing large numbers of DNA segments Next-Generation sequencing. Review throughout the text

line 372 a reference should be noted for former studies of D. melanogaster and C. elegans

lines 384 and 385 clarify this sentence 

line 406 the study by Gao et al., 2020 is on beans

lines 424-444 Please clarify these sentences and that the references correspond correctly.

lines 453-464 In my opinion you should structure your closing paragraph with clear statements based on your experimental results. 

line 590 correct the citation (year of publication and names of authors)

line 625 the year of publication should be placed after all authors.

line 631 there is a space missing after the word ant

change "form" to "from" in figure S1

Table S1 Did you do any test to check the specificity of these primers? In the supplementary material, you can add the sequences used for the design of primers

Author Response

Reviewer 1

We are very thankful to the reviewer for highlighting deficiencies and improving our manuscript

In my opinion, it is necessary to carefully review the references cited throughout the article, there are many errors throughout the entire introduction and discussion.

Ans: We carefully looked at the entire article and made all of the comments very carefully.

The scientific names of the species are written in italics, correct this throughout the text.

 Ans: Throughout the article, we make all scientific names italics.

lines 2-3 I think you should be more specific in your title because your work focused on IIS-pathway associated genes.

 Ans: We make the correction from general to specific… The longevity phenomenon is entirely controlled by the insulin signaling pathway (IIS-pathway). Both vertebrates and invertebrates have IIS-pathways that are comparable to one another.

line 24 remove "assembled" reads refers to the raw sequences provided by the sequencer before assembly.

Ans: We rewrite the sentence ….Through transcriptome analysis, we annotated 103,589,264 sequence reads, and 184,436 (7G) unigenes were assembled, GC performance was measured at 43.02%, and 64,046 sequences were reported as CDs sequences…

lines 55 and 56 To specify the references corresponding to C. elegans, the article by Giannakou and Partridge, 2007 corresponds to Drosophila.

Ans: We added the exact citations….The inhibition of upstream IIS-pathway components (daf-2 and age-1) significantly prolongs the life and health span of Caenorhabditis elegans (Kimura et al., 1997 ). On the other hand, IGF-I and IGF-II monitor growth and growth factors (GF) (Kimura et al., 1997); nutrient sensing, stress replication (Luo et al., 2020), cell functions, metabolic regulation is gradually deteriorating due to aging (Luo et al., 2020; Ros and Carrascosa, 2020).

lines 85 and 375 is next-generation sequencing (NGS).

 Ans: We correct Next-generation sequencing (NGS) into next-generation sequencing (NGS).

line 64 the article by Li et al., 2009 is about a bioinformatics tool.

Ans: Correction is done: Researchers previously reported that IIS-associated genes that prolong the lifespan of Drosophila melanogaster (Sim and Denlinger, 2013; Altintas et al., 2016), dauer state in C. elegans (Wang and Kim, 2003; Tissenbaum, 2012), and reproductives diapause in Culex pipiens mosquito (Kang et al., 2014).

linea 66 please specify a reference for Culex pipiens because Kennedy, 2008 is a review, and Tissenbaum, 2012 is in C. elegans.

Ans: We added the correct citation: .....reproductives diapause in Culex pipiens mosquito (Kang et al., 2014).

lines 66,67 and 68 correct text or reference because Carey's 2002 study was conducted on Hexagenia limbata and Ephemera simulans, and Jemielity et al., 2005, the authors discuss studies on aging in social insects.

Ans: We make the corrections…. It also indicated that squirrels live for twenty-five years, rats live for three years, and C. elegans, D. longispina, and D. melanogaster live only for a few weeks (Carey, 2002; Jemielity et al., 2005). Long-lived queens of insects, including Ephemera simulans (Carey, 2002), Pogonomyrmex owyheei (Porter and Jorgensen 1981), and Lasius niger can live up to ten years (Kramer et al., 2016).

lines 68-70 The reference for this sentence is missing.

Ans: We added the citations… (Carey, 2001; Vargo and Husseneder, 2009).

line 80   check the reference (Taniguchi et al., 2006).

Ans: We added

We followed the IIS-pathway (PI3K-Akt are the main effector pathways in IIS-signalling) in insects (Taniguchi et al., 2006; Buitenhuis and Coffer, 2009; Badisco et al., 2013; Kakanj et al., 2016) to determine the IIS-pathway in primary and secondary worker reproductives (many subterranean termite colonies consist of former worker termites or immatures that have developed into larvae and eventually become wingless reproductives to supplement the colony), and non-reproductive subterranean termites (Sonenberg and Hinnebusch, 2009; Haroon et al., 2020; Haroon et al., 2021).

lines 83 and 84 correct text or change references Mirzoyan et al., 2019 is a study in Drosophila.

Ans: We cited the correct reference...the IIS-pathway in primary and secondary worker reproductives (many subterranean termite colonies consist of former worker termites or immatures that have developed into larvae and eventually become wingless reproductives to supplement the colony), and non-reproductive subterranean termites (Sonenberg and Hinnebusch, 2009; Haroon et al., 2020; Haroon et al., 2021).

lines 85 Next-generation sequencing (NGS) 

 Ans: We correct Next-generation sequencing (NGS) into next-generation sequencing (NGS).

lines 86 the articles cited did not study the species Reticulitermes chinensis review references.

Ans: we added the correct citation….Longevity in Reticulitermes chinensis was determined through next-generation sequencing (NGS) or transcriptomic (Haroon et al., 2022).

lines 101 check font size in the word munificent.

Ans: We doubled check the word munificent and found it correct.

line 105 The analysis females x females is in different castes?

Ans: During the initial colonial foundation, we make male x females colonies and females x females colonies. To check out hybridization but no satisfactory results were found. 

line 115 remove the dot inside the parentheses.

Ans: We have removed the dot.

line 147 spinning means centrifuge?

 Ans: Yes it’s mean centrifuge…. to RNase-free centrifuge tubes (for centrifuge spinning) to separate….

line 228 same as line 24

Ans: We made the changes.

linea 154, 156, 159 and 160 It is convenient to write down the catalog number of the reagents used.

 Ans: For worldwide user’s we used the kit's name and manufacturer name.

lines 160-162 check the citation for this sentence or describe if the methodology was like the one used in that article.

Ans: We make the corrections.

line 174 here, it corresponds to note the reference number 40, Li et al., 2009.

Ans: We rewrite the reference.

line 241 I suggest changing the reference to Figure S5 at the end of the paragraph.

Ans: We added the reference Figure S5 at the end of the paragraph.

linea 252 missing letter "t" Tritrichomonas

Ans: We have corrected the spelling.

line 271 review the total sum of unigenes reported in table 2 and correct text (36,681).

Ans: Corrected. 

line 297 corrects the number 125,636

Ans: Corrected

line 258 Please check if this comment corresponds to the figure (S5)

 Ans: We already made this comment.

line 320 It is convenient to maintain the same order of the castes both in the figure and in the figure legend.

Ans: The corrections of figures and the legends were corrected.

lines 320 and 321 the text in the figure legend is contrary to the figure, insert the indication is inverted red and green.

Ans: Figures and the legends are corrected.

lines 334-337 I don't understand that Tsc2 is one of the 6 genes that they selected to analyze and two lines later they say "Tsc2 is an autosomal-dominant disease that causes epilepsy, body volume, and mutant cells"

Ans: In these lines of description, it means that the Tuberous Sclerosis Complex (TSC) is a genetic condition that is inherited in an autosomal dominant manner. It is caused by a mutation in either the TSC1 or TSC2 gene and is characterized by the development of tumours or hamartomas in many organs.

lines 339-353 I think a table with these results is better.

Ans: These data were already presented in figure number 6.

line 363 corrects the word "cates".

Ans: Corrected.

line 375 I suggest changing "or" to "with" or "from" Next-generation sequencing because the abbreviation (NGS) is used to refer to methodologies for sequencing large numbers of DNA segments Next-Generation sequencing. Review throughout the text.

Ans: Corrected.

line 372 a reference should be noted for former studies of D. melanogaster and C. elegans.

Ans: We make the corrections….Transcriptomically, we confirmed that several genes (Pdk1, akt2-a, Tsc2, mTOR, EIF4E, and RPS6) in R. chinensis had extended the lifespan like D. melanogaster (Sim and Denlinger, 2013; Altintas et al., 2016) and C. elegans (Wang and Kim, 2003).

lines 384 and 385 clarify this sentence.

Ans: We correct the entire sentences.

line 406 the study by Gao et al., 2020 is on beans

Ans: We remove the Gao et al., 2020 from these lines.

lines 424-444 Please clarify these sentences and that the references correspond correctly.

Ans: We double-check the corresponding references and found them correct.

lines 453-464 In my opinion you should structure your closing paragraph with clear statements based on your experimental results. 

Ans: We rewrite the paragraph for better understanding…..Current investigation, the hypothesized findings (RT-qPCR results) were obtained from the reproductive castes (PK, PQ, SWRK, and SWRQ), with much higher expression than in the non-reproductive castes (WM and WF). Our findings ultimately shed new insight into the maintenance and significance of biomolecular homeostasis in reproductive and nonproductive R. chinensis castes. As previously described by a number of researchers, that IIS pathway-related genes desperately promote growth factors, physiological processes, cell metabolism and survival, autophagy, fecundity rate, egg size, and follicle number. These findings imply that a relatively conserved protein in the insulin signaling system significantly prolongs or avoids age-related diseases, processes, structures, and associated pathways. Further research is required to determine the biological function of genes connected to the insulin signaling pathway using genetic tools such as RNA interference and transgenic structures in order to determine how these genes contribute to the continued existence of termites. In the end, the findings we were able to obtain new shreds of insight into the processes involved in preserving biomolecule homeostasis and its connection to remarkable longevity.

line 590 correct the citation (year of publication and names of authors).

Ans: We double-checked and corrected the corresponding citations and references.

line 625 the year of publication should be placed after all authors.

Ans: Corrections were done.

line 631 there is a space missing after the word ant.

Ans: Corrections were done.

change "form" to "from" in figure S1

Ans: This is the name of the systems/technology Illumina HiSeqTM 4000 platform (Figure S1).

Table S1 Did you do any test to check the specificity of these primers? In the supplementary material, you can add the sequences used for the design of primers.

Ans: The primers were already added to the supplementary materials.

Table S1. The six selected genes from R. chinensis and their primers were used in RT-qPCR analyses.

Gene ID

Symbol

Primers sequences

Beta-actin

Forward: CCCAACACAGCGTCTTACAA

Reverse: CAGATGTCCTCAGCTTCACG

Unigene 0082575

PdK1

Forward: TCCTCCTCCTGCTACTGCTGAAG

Reverse: CGACATATGACGGAGTAGGTGGTG

Unigene 0034890

akt2-a

Forward: CCAAGAAGTATGTCGAAAGAAGTCA

Reverse: TCTGTGAAACCATCTCCCAATTAAG

Unigene 0092210

Tsc2

Forward: AGTGGTGCTAACATGCCTGC

Reverse: ACCTTCCAGCTGCTCTGACA

Unigene 0063105

mTOR

Forward: GGCTTGAAGGGTGTTCCACA

Reverse: GCTTACCTGTAGGCGGCAAT

Unigene 0011832

EIF4E

Forward: GGATCTCGTCTTGGCCGTCATTG

Reverse: AGCAACCTCGTGAGCCACTCC

Unigene 0155613

RPS6

Forward: TCTATGACAAGCGCCTTGGAGAAG

Reverse: CTTGGACGGAGGCAGGTGGATC

Reviewer 2 Report

The English of the article needs to be further improved, and there are obvious wording problems in some places. Difficulties in reading for readers.

The way of organizing the articles needs to be reconsidered. At the same time, there are errors in the data. The team should consider re-reviewing the content of the text.

I have noticed that the literature does not correspond to the text of the article. The description of the text cannot be found in the cited articles, which makes the Articles are hard to read and would like to be reorganized.

Overall, scientifically I think it's acceptable, need to pay more attention to details.

Author Response

Reviewer 2

The English of the article needs to be further improved, and there are obvious wording problems in some places. Difficulties in reading for readers.

Ans: Dear reviewers, thank you for your comments. The manuscript is polished for English by a native speaker as well as other coauthors. We welcome if there is any specific correction required.

The way of organizing the articles needs to be reconsidered. At the same time, there are errors in the data. The team should consider re-reviewing the content of the text.

Ans: We double-checked the data where we made the corrections. Data calculation was done as well and made the corrections.

I have noticed that the literature does not correspond to the text of the article. The description of the text cannot be found in the cited articles, which makes the Articles are hard to read and would like to be reorganized.

Ans: Throughout the entire article, the citations and references were done correctly. And added new references where necessary.

Overall, scientifically I think it's acceptable, need to pay more attention to details.

Ans: We make the correction for the improvement of the articles; thank you for your valuable comments.

Round 2

Reviewer 1 Report

In this second revision I found some minor details that require your attention for the publication of your work. I wish you good luck.

Please review lines 21, 318 and 331 to unify abbreviations and description within parentheses for WF and WM

Line 256 review missing letter "t" in Tri-richomonas foetus

Change "form" to "from" in figure S1 in the sentence " extraction form termite head"

My suggestion was to add in the supplementary material the complete sequence of the unigenes
0082575 0034890, 0092210, 0063105, 0011832 and 0155613  used to design the primers, because with
 the sequence of the primers I did not find similarities in other species of the genes that you studied.
Possibly with the complete sequence of the unigenes it will be easier to verify the specificity of the amplifications.

Author Response

A point-by-point response to referees’ comment

Manuscript ID: ijms-1877527

Manuscript title: De novo transcriptome assembly and analysis of longevity genes using Subterranean termite (Reticulitermes chinensis) castes

Name of the journal: Molecular Genetics and Genomics

Author list: Haroon, Yu-Xin Li, Chen-Xu Ye, Su Jian, Ghulam Nabi, Xiao-Hong Su and Lian-Xi Xing

Dear Editor Mason Chen

Subject: Submission of revised paper mentioned above

We are grateful for the thoughtful comments of the reviewers and editor. We have carefully reviewed the comments and have revised the manuscript accordingly. Following this letter are the reviewers’ comments with our point-by-point responses for each specific comment raised by the reviewers in green font, including how and where the text was modified. The changes made in the manuscript is shown in a separate track change document and uploaded to the system with the revised version of the manuscript. We hope the revised version is now suitable for publication and look forward to hearing from you in due course.

Yours sincerely,

Nabi

Response to reviewers’ comment

Please review lines 21, 318 and 331 to unify abbreviations and description within parentheses for WF and WM.
Ans: We reviewed the

Line 21……non-reproductive (male workers “WM” and female workers “WF”) castes of the subterranean termite Reticulitermes chinensis.

Line 318…. primary king (PK), primary queen (PQ), secondary worker reproductive king (SWRK), secondary worker reproductive queen (SWRQ), workers male (WM) and female workers (WF).

Line 331….Primary king (PK), primary queen (PQ), secondary worker reproductive king (SWRK), secondary worker reproductive queen (SWRQ), male workers (WM) and female workers (WF).

Line 256 review missing letter "t" in Tri-richomonas foetus
Ans: We made the corrections….. Tri-trichomonas foetus.

Change "form" to "from" in figure S1 in the sentence " extraction form termite head".

Ans:  Corrected

My suggestion was to add in the supplementary material the complete sequence of the unigenes 0082575 0034890, 0092210, 0063105, 0011832 and 0155613  used to design the primers, because with the sequence of the primers I did not find similarities in other species of the genes that you studied. Possibly with the complete sequence of the unigenes it will be easier to verify the specificity of the amplifications.

Ans: corrected accordingly

>Unigene0082575  +  nr

ACCAGAAGTGTTAAATAACGGAAAGATTTCGCCTGCAAGTGATCTCTGGAGCTTCGGTTG

TATTTTGTTTTCCATTATCGACGGAAAGCCTCCCTTTTATACAGGGAACATGATGAATAC

ATTTGCTGCTATCGAGAAAGGAACCATTGACTTTCCACCGACTTTCTTGCCACAAGCGAA

AGACCTTGTTCAGAAACTTTTGAAGCTCGAACCCTCTGAACGATTGGGTTTCAATGAGTA

CCCGACCAATTACAAGCCCATTCGAGACCATCCATTCTTTTCTGGATTGGATTGGGAGAC

TCTTCCACTTGAAGAAGTTCCTCCTTTCGATGACGATGAACCATTTATCCTTCCTCCTCC

TGCTACTGCTGAAGTACCTTCATCCAAACCTTCTTCGAAACCTCCTCCTTCTGATTCTCC

TCCAGATAGTTCTCCCTCTCCGAAAGCCTCTTCACCAGCTCCTTCTTCTCCCGCTCCTGC

ACCACCTACTCCGTCATATGTCGTTGATCCTGCTAAGTATCTCAGTCTCATCAATTCTCC

TAACCTGCTTGATGAATCAAAACAATTTCTGATTCAAGACGAGGTAATTATCTATCAAGG

GCTTCTGTGGAAGCGTGTTGGACTTTCAAATAAGGAAAGGTTGTTTGTGATAACGACAAG

GCCTCGAATCTTCTACTGGGACTTGAAGGGCAAAAAATTCAAAGGAGAGATTCCTCTTTC

AAAAGAACTTGAGGTTTCCATTGAAAAAGGAGGAAAGTTTATTCTCGAAGTGCCCGGCCG

TGTGTATAAGCTCG

>Unigene0034890  +  nr

GCCTTTTTACTGCGAAAACCAGAATCAAATGTTCAAAGATATCCAATCAAAGCAAGTCAA

GTATCCAAGAAGTATGTCGAAAGAAGTCAAAGACTTGATTTCGAGATTTCTTGTCCGTGA

CATCTCTCAGAGAATTGGAGCTGGTCCAGAAGATTATGAAGAAGTTAAGAGACATCCTTG

GTTTAGTGATCTTAATTGGGAGATGGTTTCACAGAAAAAGATTACACCTGAATGGAAGCC

TG

>Unigene0092210  275  1060  Tuberin [Zootermopsis nevadensis]

ATGAGTGCGAAGGATAAGGAAAACAAAACTTTTCACGAGAAACTGAAGCAGTTTTTCAGA

ATAAACAAAGGTGGAACAGGTAACTTGAAAGGCAGAGTGGATTTTGCTTTGACTCAAGAC

ATTGAGAAAGACCTCAGCCCTGAAAACCCAGTGACTCATCGCGTGAAGGTTATAAAGGAA

CTGAGTGAAGCCGTTCTTAAGAACCGTCTTGAGGATAATTCAATTGAGAAATTATGGGCA

TGTCTTCAAGATTTGTTACATCGTGAAGTACTAAAAGAACATCGCCATCTTGCTTTTTAC

TTTTTCCGTTGCCTTGTACAGGGACAGTATGACAAACTAGGGCTTATGAGGGTTCACTTC

TTCAGAATCATTAAGACTCATGATATTCCAGAAGATGTAGCACCAAGATTTGAGCTGTTG

CAAAGTCTGACAGACAACGGGAAAGACATTCTGTACTTTGAAGAAGAGGTGGGGCCTTTT

CTACTTTACTGGATGCCAGCTGTGATTGGTGTTGCTCGCACAAAGGAATTCCTCTCCATG

CTTGTCAATGTTATCAAGTTTAATGCTGCTTATGTTGATGAGGATGTTATTTCAGGTCTT

GTTCAGAATACATGTTTCCTGTGTTGTTGGAGCAACTCGGAGGAAGTGGTGCTAACATGC

CTGCAAGTGCTGGACACAGTTGTCTGTTACAGTACCTTGCCTTCTGACTCTCTGCATACC

TTTATCAGTGCCCTTTGTCGAACTGTCAATGTAGAAGCGTATTGTCAGAGCAGCTGGAAG

GTGATG

>Unigene0063105  3  665  PIKK family atypical protein kinase [Trichomonas vaginalis G3]

AATCAAGAGTCTCTTCATCAAGCTTGGCAGTTCTATATTAATCTATATCGTCAAGTGAAA

ATGATTGTTCTAAACTTAATGACAATTCCTCTTGCTGAAGCTTCTCCTAAACTGTCCTCA

GTTTTAAGCTTTTCTTTGTCTGTCCCTGGAACTTACCATCACAATAGTAATATCATTACG

ATTCAATCATTTCAACCTCTTTTAAAAGTTTTACCTTCGAAGCAAAGACCCAGAAGAATG

GGAATTATTGGAAGCGATGGAAATTCTTATACATTTTTGTTGAAAGCAAGAGAAGACACT

CGTCTTGATGAGCGAGTTATGCAGTTATTCACTTTCTTAACTTCACTTGTGAACAGTTCA

GCAATTCCAATGAAAAACAAGTTAACGATTACAACTTATAATGTCATTCCTTTAACACAT

GAAGTTGGATTAATTGGTTGGCTTGAAGGGTGTTCCACAATTTATGATCTTATCTTGGAA

CATCGAAAGAAAAATTCAATCGCAACTAAAAAGGAATATGAATATGCAATTAAAAAATAT

CCAACTTATAATCAATTGCCGCCTACAGGTAAGCTCAAGGCATTCAGAGAATCCTTAAAT

GAAACCAAAGGAGATGATTTAAAACAATTTCTCTTCAAGTTTTCAACAGATTCCTCTAAT

TGG

>Unigene0082575  +  nr

ACCAGAAGTGTTAAATAACGGAAAGATTTCGCCTGCAAGTGATCTCTGGAGCTTCGGTTG

TATTTTGTTTTCCATTATCGACGGAAAGCCTCCCTTTTATACAGGGAACATGATGAATAC

ATTTGCTGCTATCGAGAAAGGAACCATTGACTTTCCACCGACTTTCTTGCCACAAGCGAA

AGACCTTGTTCAGAAACTTTTGAAGCTCGAACCCTCTGAACGATTGGGTTTCAATGAGTA

CCCGACCAATTACAAGCCCATTCGAGACCATCCATTCTTTTCTGGATTGGATTGGGAGAC

TCTTCCACTTGAAGAAGTTCCTCCTTTCGATGACGATGAACCATTTATCCTTCCTCCTCC

TGCTACTGCTGAAGTACCTTCATCCAAACCTTCTTCGAAACCTCCTCCTTCTGATTCTCC

TCCAGATAGTTCTCCCTCTCCGAAAGCCTCTTCACCAGCTCCTTCTTCTCCCGCTCCTGC

ACCACCTACTCCGTCATATGTCGTTGATCCTGCTAAGTATCTCAGTCTCATCAATTCTCC

TAACCTGCTTGATGAATCAAAACAATTTCTGATTCAAGACGAGGTAATTATCTATCAAGG

GCTTCTGTGGAAGCGTGTTGGACTTTCAAATAAGGAAAGGTTGTTTGTGATAACGACAAG

GCCTCGAATCTTCTACTGGGACTTGAAGGGCAAAAAATTCAAAGGAGAGATTCCTCTTTC

AAAAGAACTTGAGGTTTCCATTGAAAAAGGAGGAAAGTTTATTCTCGAAGTGCCCGGCCG

TGTGTATAAGCTCG

>Unigene0155613  -  nr

CCGAAAACGGGACACAGATCTGCGTCGAAGTTACTGAACAGCAGCAGCTCGCGAACCTCT

ATGACAAGCGCCTTGGAGAAGATATTGAAGGAGAGAAGCTCGGGGATCAATTCGCAGGTT

ATGTTTTTCGCCTTGGTGGAGGCTTTGACAAGGAAGGTTTTCCAATGAAACCAGGTGTTT

TCACTCCTCGTCGAGTTCGTCTTCTCTTAAAAAAGGGATCCACCTGCTTCCGTCCAAGAG

TTAACGGAGAAAGGAAAAGAAAGTCTGTTCGTGGCTGTATCATTTCTTCTGAGATTTCCG

CGCTTCACATTATTGTCATTCAGAAAGGTCCTGGAGAGATTCCAGGTCTCACTGATCTCC

ATGTCCCAAGATTGTATAGCCCCAAGCGTGCTTCCACACTGAAGAAGATGTTTGATCTTA

AGACAAATGAAGAGGTAGCTAATGCAGCTATTCAAAGGCAGACTAAGTCAGGTAGGTTTG

TGAAACCAAAGATCCAGAGACTAATTACTCCTCGCAGACTTCAACGAAAGAAGAAAGAAC

AACAAGAAAG
